# The Effect of *Aloe vera* on Skin and Its Commensals: Contribution of Acemannan in Curing Acne Caused by *Propionibacterium acnes*

**DOI:** 10.3390/microorganisms12102070

**Published:** 2024-10-16

**Authors:** Suraj Pal, Mayank Raj, Medha Singh, Kumar Saurav, Chetan Paliwal, Subhasish Saha, Anil Kumar Sharma, Manoj Singh

**Affiliations:** 1Department of Bio-Science & Technology, Maharishi Markandeshwar (Deemed to be University), Mullana-Ambala 133207, Haryana, India; palstudy123@gmail.com (S.P.);; 2Laboratory of Algal Biotechnology-Centre Algatech, Institute of Microbiology of the Czech Academy of Sciences, 37901 Třeboň, Czech Republic; sauravverma17@gmail.com (K.S.); saha@alga.cz (S.S.); 3Laboratory of Photosynthesis-Centre Algatech, Institute of Microbiology of the Czech Academy of Sciences, 37901 Třeboň, Czech Republic; paliwal@alga.cz; 4Again Bio, 2860 Søborg, Denmark; 5Department of Biotechnology, Amity University, Mohali 140306, Punjab, India; anilbiotech18@gmail.com

**Keywords:** *Aloe vera*, *Propionibacterium acnes*, opportunistic pathogen, antimicrobial, superoxide dismutase, peroxidase, acemannan

## Abstract

*Aloe vera* is one of the most significant therapeutical plant species that belongs to the family Liliaceae. *Aloe vera* is composed of a high amount of water, with the remainder being dry matter. The dry matter contains a lot of bioactive compounds like carbohydrates, fats, and enzymes, with various therapeutic and antimicrobial properties. It can enhance the proliferation of cells and prevent cell damage by anti-oxidative properties (stimulating the secretion of superoxide dismutase and peroxidase). Human skin is colonized by microbes like fungi (*Candida albicans*), bacteria (*Propionibacterium acnes*, *Staphylococcus aureus*), and mites. These commensals are responsible for skin characteristics such as acidic pH, the pungent smell of sweat, etc. Human fetuses lack skin microbiota, and their skin is colonized after birth. Commensals present on the skin have a crucial role in training the human immune system against other pathogenic microbes. *Propionibacterium acnes* act as an opportunistic pathogen when the balance between the commensals is disturbed. We also emphasize the recent progress in identifying the aloe metabolite biosynthesis pathways and the associated enzyme machinery. The hyperproliferation of *Propionibacterium acnes* causes acne, and acemannan plays a significant role in its cure. Hence, we need to consider a new treatment approach based on the root cause of this dysbiosis.

## 1. Introduction

*Aloe vera* is a tropical, drought-resistant plant that belongs to the family Liliaceae. *Aloe barbadensis* is now referred to as *Aloe vera* by taxonomists. *Aloe vera* is derived from the Arabic word “Allaeh”, which means “shining bitter substances”, and “Vera” is a Latin word that means “True”. Northern USA and the Islands of Barbados have the biggest *Aloe vera* plants with yellow and brown milk-filled leaves [1]. This milk makes up the bulk of the leaves and is covered by large-walled mesophilic cells. Some species of *Aloe vera* have been found to be toxic to the environment. However, some species have been confirmed to possess therapeutic properties [2]. Aloe gel is composed of 96% water, with the remaining 4% being dry matter that includes dietary fiber (73.35%), ash (16.88%), protein (6.86%), fat (2.91%), and ascorbic acid (0.004%). The percentage composition of dry matter present in the *Aloe vera* is presented in Figure 1.

**Figure 1 microorganisms-12-02070-f001:**
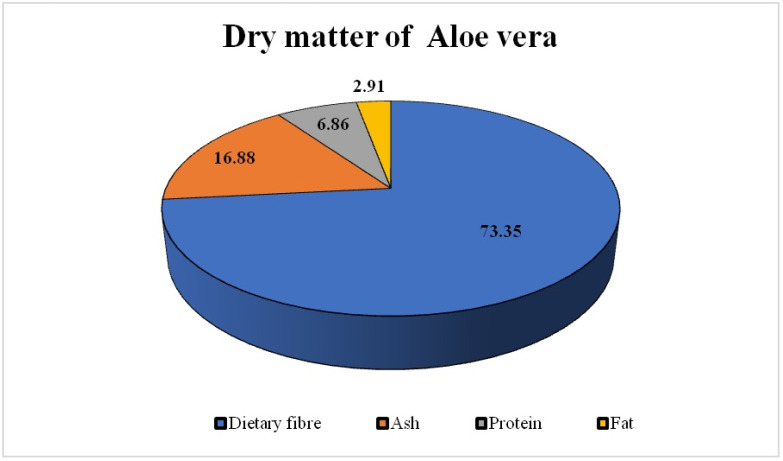
Dry matter composition. *Aloe vera* contains a large quantity of organic compounds including polysaccharides, enzymes, salicylic acid, tannins, fatty acids, vitamins, lectins, flavonoids, minerals (electrolytes), and proteins. Examples of these compounds are given in Table 1. Glucomannan and hemicellulose fibers are present in the aloe and play a crucial role in enhancing cellular proliferation. The anthracene derivative of quinone, also known as anthraquinones, is a laxative agent present in the aloe. Aloe-emodin and aloin are the two basic anthraquinones in aloe. Aloe-emodin is provided in powdered form as a laxative. The antioxidant (in vitro) properties of *Aloe vera* gel have been confirmed in some species of aloe including *Aloe ferox*, *Aloe arborescens*, *Aloe greatheadii var Davyana*, *Aloe harlana*, *Aloe marlothii*, *Aloe melanacantha*, and *Aloe saponaria*.

The antioxidant properties of *Aloe ferox* were also determined using ORAC (Oxygen Radical Absorbance Capacity) and FRAP (Fluorescence Recovery After Photobleaching) analysis [3]. *Aloe vera* also has various therapeutic effects. Around 30 species of *Aloe vera* have been confirmed to have therapeutic properties. Hemicellulose fibers and glucomannan present in the *Aloe vera* gel help enhance cell proliferation, resulting in effective wound healing in plants. Mucopolysaccharides have water retention capabilities and help keep skin moisturized [4]. Acemannan is a strong plant-based immunomodulator that controls various immunological reactions and has various pharmacological roles with some other bioactive compounds. Some compounds that have pharmacological impacts on various sites of the human body are represented in Table 2. These compounds are also produced by the human body, but only starting in adolescence, and their levels can decrease due to stress, which can lead to swelling, digestion problems, nerve root pain, and various infections. Hypoxia and poisoning are also related symptoms. Acemannan also maintains the normal flow of lymph and gaseous exchange in the alveolar sac [5]. Polysaccharides are present in abundance in aloe gel and provide a range of health benefits, including hypo-glycemic (anti-diabetic) and gastro-protective effects. They also exhibit anticancer, immunomodulatory, anti-inflammatory, and antifungal activity. Additionally, they improve absorption in the intestine and enhance skin permeation [6]. A study illustrated the effect of *Aloe vera* on the permeation of skin with the help of ketoprofen as a marker. This study was performed on female abdominal skin in Franz diffusion cells, focusing on transfer of ketoprofen in the epidermis and corneum of the skin using tape-striping technology. The results showed an elevated permeation-enhancing effect (enhancement ratio = 2.551) of ketoprofen by *Aloe vera* gel [7]. Pepsin is a hydrolytic enzyme present in the human stomach.

Pepsin can work optimally in acidic pH; however, an acidic pH can also damage the lining of the stomach. Glycosaminoglycans activate the mucosa, stimulate the secretion of mucous, and reduce stomach irritation. They can also prevent cellulite formation. Dihydrochloride can induce oxidative stress and cause cell death in the kidney’s epithelial cells. *Aloe vera* also provides protection against dihydrochloride [8]. Peroxidase can eliminate the hydrogen peroxide on the skin surface and reduce the chances of cellular damage. Nwajo stated that elevated sugar levels in the blood elevate oxidative activity, based on a study conducted on diabetic rats. He used the leaves of *Aloe barbadensis*, which elevated the level of superoxide dismutase activity and restricted lipid peroxidation [9]. After the treatment, elevated levels of some other antioxidative enzymes, such as catalase, glutathione peroxidase, and glutathione-S-transferase, were observed in the liver and kidneys of rats [10]. The authors also showed that ferric ions in a solution and polyphenolic components of solutions have a very strong ability to eliminate free radicals based on FRAP and ORAC analysis. These chemicals may be used against oxidative-stress-related diseases. Aloe epidermis extracts are more able to scavenge free radicals than aloe gel, as confirmed by Monniruzzaman et al. and Lopez et al. Their findings showed that the alcohol extracts from the aloe epidermis contain a higher DPPH radical-scavenging tendency and a higher tendency to reduce Fe (II) to Fe (III) [11,12]. However, the aloe gel has a greater radical-scavenging tendency than whole leaf extract. Moniruzzaman also showed that DPPH radical-scavenging and ferric-reducing tendencies were highly correlated with the amounts of phenolic compounds and flavonoids present in ethanol aloe extract [13]. *Aloe vera* gel can increase ketoprofen permeation through dermatomes (specific areas of skin that are connected to the nerve root of the spine).

## 2. Human Skin and Its Commensals

The skin is a physical barrier and has a surface area of 2 m^2^. It is made up of three layers (the epidermis, dermis, and hypodermis) that help the body immunologically by preventing the entry of various pathogens in multiple ways [14]. Human skin directly interacts with the outer environment and is colonized by a wide variety of microbes, including bacteria, fungi, viruses, and even mites. The skin provides a variety of microenvironments due to its thickness, folds, mesh of hair follicles, temperature, and acidity. It also acts as a physical barrier and restricts the penetration of microbes and harmful toxins to itself. Most of the skin surface is covered with fluid (sweat)-secreting glands (also called apocrine glands). The sweat primarily contains water and salt. Sweat plays significant roles in skin acidification and thermoregulation, which help to prevent the growth of microbes and cool the skin through the evaporation of its water content, respectively. Acidic pH inhibits the growth of *Staphylococcus pyogenes* and *Staphylococcus aureus* [15,16,17]. Our skin holds various antimicrobial proteins, acidic pH, and free radicals that help inhibit microbial colonization [18]. The pungent odor of sweat indicates bacterial colonization and results from the breakdown of apocrine secretions by bacterial growth. Sebum is a lipid-rich hydrophobic secretion produced by the sebaceous glands present in hair follicles. Sebum forms a layer on the skin that provides an antibacterial environment. However, the face and chest have a higher density of sebaceous glands, which support lipophilic organisms such as *Malassezia* spp. As sebaceous glands are present in the hair follicles, they are relatively anoxic, promoting the growth of facultative anaerobic microbes like *Propionibacterium acnes*.

There is a relatively lower number of organisms found in the arm and leg regions based on culture-based methods due to greater fluctuations in temperature and moisture. Skin commensals play a significant role in developing our immune system to effectively resist pathogenic infections [19]. *Candida albicans* is another commensal found in all human populations as part of the normal flora, according to molecular detection methods, and is present in the oropharyngeal, gastrointestinal cavity, and vaginal tracts of healthy humans. It is associated with various morphological forms like blastospores, pseudohyphae, and hyphae [20]. Commensals are generally associated with certain pathogenic activities. As we know, the skin provides a wide range of microenvironments on its surface and supports a wide variety of commensals. These commensals are helpful for us in developing our immune system to fight against other pathogenic organisms, especially B-lymphocytes. The host’s cutaneous innate and adaptive immune system can modulate the skin microbiome. However, any dysregulation or disbalance can lead to opportunistic pathogenic infections [21]. Some examples of such opportunistic pathogens are shown in Table 3.

The interaction of the host with its skin commensals also has a significant role in developing the immune system in infants. The importance of initial microbial colonization during birth has been illustrated by a mouse experiment in which neonatal mice were exposed to *Staphylococcus epidermidis* (coagulase-negative staphylococci) during early life. As a result, mice developed a large number of Treg cells (specific for *Staphylococcus epidermidis*), which helps prevent further inflammation during recolonization. However, the absence of this exposure in early life may reduce inflammatory prevention later on [22]. There are some factors associated with the host that influence the quantity and variability of normal skin flora. These factors are sex, location, and age. Age has an effect on the variability of the microenvironment present on the skin surface of the host [23].

**Table 3 microorganisms-12-02070-t003:** Some skin microbiota and skin diseases associated with them.

Microorganism	Associated Skin Diseases
*Staphylococcus aureus*	Impetigo, staphylococcal scalded skin syndrome
*Staphylococcus warneri*	Botryomycosis
*Streptococcus pyogenes*	Erysipelas, cellulitis, impetigo
*Cutibacterium acnes*	Acne vulgaris
*Corynebacterium* spp.	Erythrasma, pitted keratolysis, trichomycosisaxillaris
*Acinetobacter johnsonii*	Green nail syndrome, hot tub folliculitis, toe web infection
*Pseudomonas aeruginosa*	Subcutaneous nodules, pustules, ecthyma gangrenosum,
*Candida albicans*	Candidiasis, intertrigo, tinea versicolor, athlete’s foot
*Trichosporoncutaneum*	White piedra
*Rhodotorula rubra*	Fungal oesophagitis

As we know, in the gestational period, the fetus’ skin is sterile due to it having no direct contact with microbes. The growth of commensals can be noticed just after birth, due to the contact with the external environment either at the time of vaginal delivery or cesarean delivery [24]. The anatomical and physiological changes between males and females, including differences in sebum, hormone, and sweat production, affecting microbial colonization in their skin microenvironments. Another significant factor is the environment in which the person is living and their lifestyle. The external environment acts as a source for a wide variety of microbes and also provides basic factors like temperature, pH, moisture, etc. Elements like the natural environment, cosmetics, soap, moisturizers, and other hygienic products can also contribute to diversity of an individual’s microbiota [25,26].

However, how these products affect skin commensals is still not known. The quantitative culture technique showed us that bacteria grow more in high temperature and high humidity compared to high temperature and low humidity in the areas of axillary vaults and feet. It has also been evidenced that low temperature with low humidity is responsible for higher colonization of Gram-negative bacteria on the feet [27]. The diversity of the skin microbiome also depends on factors beyond the skin’s characteristics. In a study, it was found that the proteobacteria and actinobacteria were found in higher and lesser amounts, respectively, in the guts of individuals with mild to severe acne in comparison to healthy persons. This study was conducted on 31 acne patients. Another study also showed the impact of a Western diet on the growth of acne [28]. The intake of dairy products like saturated fats, simple carbohydrates, and dairy products somehow activates some metabolic reactions and helps in the formation of acne. It was shown that the intake of omega-3 in supplements can reduce lesions in acne patients. Any bacterial dysbiosis in the gut results in increased intestinal permeability. Inflammatory chemicals like lipopolysaccharide endotoxins can be released in the circulation and may cause inflammation of the skin [29].

Some *Aloe vera* species, such as species from Frangula and Rhamnus, are rich in anthraquinone. These species have well known for their antimicrobial properties against *P. aeruginosa*, *E. coli*, *A. niger*, *Candida albicans*, *S. aureus*, etc. The effects of anthraquinone against malaria have also been investigated [30]. Acemannan has proven to have bactericidal, virucidal, and fungicidal activities by activating the macrophage cells to fight against pathogenic microbes entering the host. It can restrict the intestinal absorption of toxins into the circulatory system. It also shows bacteriostatic effects on *Streptococcus viridans* and antioxidative and cytotoxic effects on squamous epithelial cells in lung cancer [31].

## 3. *Propionibacterium acnes* Pathogenesis and the Role of Acemannan in Acne Treatment

*Propionibacterium acnes* is a Gram-positive, anaerobic, aerotolerant, and non-spore-forming bacterium of the phylum Actinobacteria. It is situated at sebaceous sites on the skin, where it attaches to keratinocytes [32]. Genome sequencing of *P. acnes* shows the presence of the *gehA* gene that encodes lipases. These will further help in the adherence of the bacteria on the skin. *P. acnes* can also produce hyaluronidase and protease, which can damage the sebaceous glands. This damage can activate the classical and alternative complement pathways. This pathway may induce the production of cytokines and neutrophil chemotactic factors [33]. Figure 2 represents the mechanism of acne development. *P. acnes* participate in the inflammatory responses of the skin by activating the innate immune system. *P. acnes* phenotype IA was observed to have more virulence properties in acne than in healthy humans. Therefore, *P. acnes* proliferation in the pilosebaceous follicle is the main factor in the development of acne. These inflammatory responses include high sebum production and hardening of the skin tissues due to abnormal behavior shown by the keratinocytes and hyperproliferation of the hair follicles. Not only does *P. acnes* contribute to the development of acne, but various other factors, such as stress, hormones, family history, etc., play a role [34,35,36]. Acemannan in *Aloe vera* plays a significant role in the first line of defense by increasing the viability of macrophages by targeting some signaling pathways (GSK-3beta/PI3K/Akt). Macrophages, T-lymphocytes, and dendritic cells show high expression of CD40, CD54, MHC-II, B7-1, and b7-2. They can promote T-cell proliferation and increase T-cell cytotoxicity in a dose-dependent manner [37]. Acemannan acts as a modulator of the immune system by increasing macrophage cytokine production, expression of surface markers, and nitric oxide release (toxic defense system against pathogens).It makes the cell membrane of skin cells tighten and creates a strong barrier to restrict pathogenic microbes. Aloe extracts have also been shown to enhance both humoral and cell-mediated immune responses.

*Aloe vera* extract can also suppress delayed hypersensitivity reactions in mouse model experiments [38]. Because of the above actions of acemannan against *Propionibacterium acnes,* external use of *Aloe vera* on the skin can reduce the hyperproliferation of *P. acnes* and cure acne. Acemannan’s anti-inflammatory properties have been demonstrated in various models, including inflammatory cells and animal studies, with the mechanisms of its action elucidated (see Figure 3). Inflammation triggers the process of phosphorylation in mitogen-activated protein kinases (MAPKs) such as c-Jun NH2-terminal kinase (JNK) and extracellular signal-related kinase (ERK). This also stimulates the production of inflammatory genes such as nuclear factor-kappa B (NF-κB) [38]. Aloin exerts anti-inflammatory action in peritoneal macrophages by dose-dependently decreasing the phosphorylation of ERK and JNK. Acemannan has the ability to inhibit pro-inflammatory cytokines such as tumor necrosis factor (TNF)-α, interleukin (IL)-6, and IL-1β. Aloin is a powerful substance that activates heme oxygenase-1 (HO-1), a protein that regulates immunological function and reduces inflammation [39].

Acemannan also has the capacity to stimulate the production of HO-1 and decrease inflammation in human umbilical vein endothelial cells that have been triggered by LPS, as well as in mouse tissues that have been injected with LPS. Additionally, it suppresses the production of iNOS, NO, COX2, and ILs. Acemannan exerts these effects by suppressing NF-κB activity and the phosphorylation of STAT-1. Acemannan inhibits the activation of JAK1-STAT1/3 and the generation of reactive oxygen species (ROS) in RAW264.7 cells, thus preventing inflammation. Acemannan and LY294002 are structurally related, with acemannan being similar in structure to the synthetic PI3K/AKT inhibitor LY294002. LY294002 is a chemical that has anti-inflammatory properties and is derived from the flavonoid quercetin [40]. Therefore, it demonstrates a comparable anti-inflammatory mechanism by activating the ROS-mediated PI3K/AKT/NF-κB pathway, similar to the effects of LY294002. High Mobility Group Box 1 (HMGB1) is a pivotal proinflammatory cytokine that is intricately associated with sepsis. Acemannan limits the secretion of HMGB1 and improves the survival of septic mice in HUVECs generated by LPS. Acemannan enhances the deacetylation of High Mobility Group Box 1, boosts the activity of sirtuin 1, and triggers the activation of PI3K/Nrf2/HO-1 signaling pathways. Previous investigations have indicated that polyphosphate (PolyP) produced from human endothelial cells is a pro-inflammatory mediator. Acemannan suppressed the activation of NF-κB caused by PolyP, as well as the production of TNF-α and IL-6. Additionally, it attenuated the septic response induced by PolyP in mice [41]. Acemannan has the ability to block the transforming growth factor β-induced protein that is caused by lipopolysaccharide (LPS), thus suppressing sepsis in mice. Acemannan’s ability to suppress the TGFBIp signaling pathway makes it a promising candidate for the treatment of diverse vascular inflammatory conditions [42].

## 4. Acemannan Application in Tissue Engineering

Clinical case reports have investigated the application of acemannan in the development of oral and maxillofacial tissues, including teeth, skin, mucosa, and alveolar bone. While these findings have opened up new possibilities, they still have some restrictions. On the one hand, tissue regeneration is not extremely efficient. Tissue regeneration engineering consists of three fundamental elements: progenitor cells, cytokines, and scaffold materials. Growth factors play a vital role in the repair process by accelerating and improving the repair outcome. Currently, there is a lack of research on the concurrent application of growth factors and acemannan. Future researchers can improve their studies by using growth factors such as bone morphogenetic proteins (BMPs) and transforming growth factor beta (TGF-β) [43]. This will enable them to examine the collective influence of these factors on the efficacy of autologous chondrocyte implantation in tissue defect restoration. Nevertheless, the many forms of repair tissues do have their own limitations. The oral and maxillofacial regions house a crucial joint structure called the temporomandibular joint, which has posed a persistent problem for those afflicted with temporomandibular joint disorders. A significant consequence of the advancement of a disease is the degradation of the cartilage in the temporomandibular joint [44].

At present, there is a shortage of a comprehensive therapeutic drug that can effectively fix defects in articular cartilage. The authors propose that including growth factors for cartilage repair could potentially improve the healing of articular cartilage defects, based on the observed beneficial effects of acemannan on alveolar bone regeneration. This would represent significant progress and constitutes a highly valuable topic of study. The influence of the drug’s application form on its therapeutic effectiveness is generally acknowledged. Hence, further inquiry is necessary to ascertain the most suitable format for the use of acemannan. Presently, the most often employed varieties are spongy, granular, and gel-activated carbon, each offering unique characteristics. Sponge and granular treatments have the ability to occupy tissue defects, but their durability is insufficient, making them unsuitable for repairing damage in high-stress locations [45]. The gel drug has high mobility but has a restricted duration of effect, requiring repeated administration. There is currently no comprehensive medicine available to address skin, mucosal, and significant bone problems. The field of acemannan materials focuses on the production of end products such patches or sprays used to cure small mucosal ulcers and skin defects, as well as the creation of dressings for the recovery of significant skin injury, among other uses. In summary, the polysaccharides present in plants play important roles in the human body, and acemannan produced from *Aloe vera* has great potential for scientific research. We expect that cooperation among researchers from many nations will result in notable progress in the fields of science, biology, chemistry, medicine, and pharmacology through the utilization of acemannan as an innovative biological material in the upcoming years [46,47].

## 5. Conclusions

As we have already seen, *Aloe vera* has therapeutic properties that impact microbial growth, along with the benefits of some secondary metabolites. We tried to understand the skin structure, composition, and various microbiomes concerning the skin locations, like underarms, face, chest, etc. *Propionibacterium acnes* is a Gram-positive, anaerobic bacterium that generally inhabits the sebaceous gland and utilizes the sebaceous secretions as its nutrition. Dysbiosis of skin commensals and hyperproliferation of *Propionibacterium acnes* can cause acne, ranging from mild to severe. Acemannan is one of the secondary metabolites present in *Aloe vera* that exhibits immunoregulatory actions on immune cells like macrophages and T-cells by stimulating pathways such as GSK-3beta/PI3K/Akt and suppressing harmful oxidative species like NO. Ultimately, acemannan plays a critical role in the treatment of acne by enhancing the action of the immune system against *Propionibacterium acnes* and could represent a new avenue for the clinical and wellness industries.

## Figures and Tables

**Figure 2 microorganisms-12-02070-f002:**
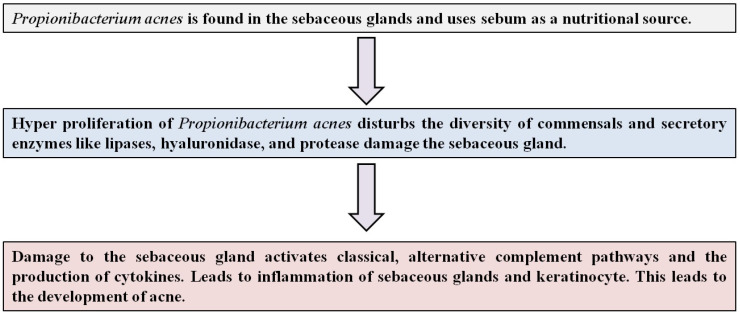
Mechanism of acne development by *Propionibacterium acnes*.

**Figure 3 microorganisms-12-02070-f003:**
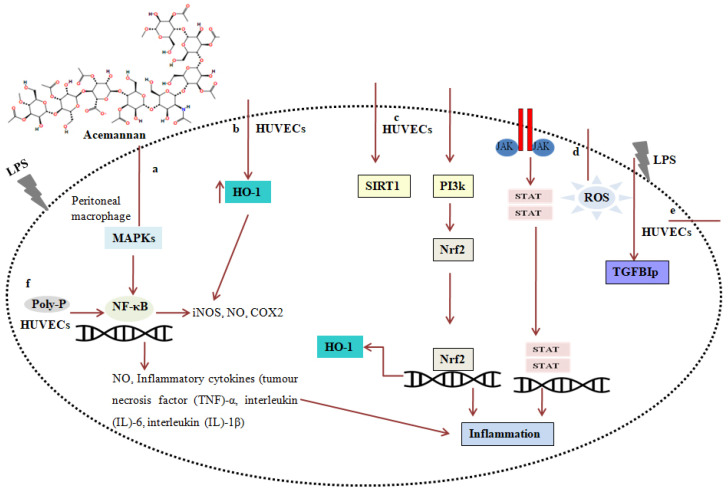
Anti-inflammatory mechanism of acemannan. (**a**) In peritoneal macrophage immune cells aloin inhibit the LPS-induced phosphorylation of ERK and JNK and inhibits secretion of Nitrogen oxide (NO) and cytokines of tumor necrosis factor (TNF)-α, interleukin (IL)-6, and interleukin (IL)-1β. (**b**) Aloin can activate the Nrf2/HO-1 defense pathway, which can enhance the expression of HO-1and reduced inflammation in LPS-activated HUVECs by inhibiting NF-κB activity and the expression of iNOS, NO, COX2, and ILs. (**c**) Aloin significantly reduce HMGB1 release in LPS via SIRT1-mediated HMGB1 deacetylation. (**d**) Aloin drastically reduces LPS-stimulated inflammation pathway in cells by activation of ROS formation and hindering JAK1-STAT1/3. (**e**) Aloin mediates inhibition of inflammation associated with sepsis by affecting the release of growth factors in LPS induced HUVECs. (**f**) Aloin inhibiting necrosis factor, TNF-α and IL-6 in human endothelial cells.

**Table 1 microorganisms-12-02070-t001:** Biochemical composition of *Aloe vera*.

Type of Macromolecules	Example of Macromolecule
Carbohydrates and its derivatives	Mannose, glucose, rhamnose, xylan, pectic derivatives, cellulose, arabinogalacton, acetylated glucomannan, acetylated mannan and pure mannan, heparin, and hyaluronic acid.
Vitamins	Thiamine, riboflavin, pyridoxin, cobalamin, ascorbic acid, folic acid, and alpha-tocopherol.
Electrolytes or Minerals	Calcium, chlorine, chromium, copper, iron, magnesium, manganese, phosphorous, potassium, sodium, and zinc.
Hormones	Auxin and gibberellin.
Enzymes	Amylase, alkaline phosphatase, carboxypeptidase, catalase, cyclooxygenase, lipase, oxidase, superoxide dismutase, bradykinase, creatine phosphokinase, and peroxidase.
Proteins	Lectins.
Amino acids	Alanine, arginine, aspartic acid, glutamic acid, glycine, histidine, hydroxyproline, isoleucine, lysine, methionine, proline, phenylalanine, threonine, tyrosine, and valine.
Lipids	Arachidonic acid, gamma-linolic acid, sterols (campesterol, cholesterol, beta-sitosterol), triterpenoids, and triglycerides.

**Table 2 microorganisms-12-02070-t002:** Pharmacological effects of some bioactive compounds present in *Aloe vera*.

Pharmacological Effects or Properties	Involved Bioactive Compounds
Digestive disease protection	Acemannan
Antidiabetic	Aloe-emodin, Aloin
Anti-cancer	Aloe-emodin, Aloin, Aloesin, Emodin
Bone protection	Aloe-emodin, Aloin
Cardioprotective property	Aloe-emodin
Antimicrobial and prebiotic effects	Acemannan, Aloe-emodin
Skin protection (inflammation)	Aloe-emodin, Aloin, Aloesin, Emodin

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
