# Peer review of "The Effect of *Aloe vera* on Skin and Its Commensals: Contribution of Acemannan in Curing Acne Caused by *Propionibacterium acnes"

_microorganisms, 2024, doi:10.3390/microorganisms12102070_

Round 1
Reviewer 1 Report
Comments and Suggestions for Authors
Pal and co-workers report the role of aloe vera in the treatment of acne, targeting Propionibacterium acnes. This review manuscript is written in a good manner and it is well organised. Therefore, the manuscript can be accepted for publication in microorganisms after the following minor revisions.
1. The headings of sections should be consistently written in bold throughout the manuscript.
2. Section 2 should be in paragraph format.
Author Response
1st Reviewer comments
Comments and Suggestions for Authors
Pal and co-workers report the role of aloe vera in the treatment of acne, targeting Propionibacterium acnes. This review manuscript is written in a good manner and it is well organized. Therefore, the manuscript can be accepted for publication in microorganisms after the following minor revisions.
- The headings of sections should be consistently written in bold throughout the manuscript.
Answer: The correction has been made in the manuscript.
- Section 2 should be in paragraph format.
Answer: The section two has been sub divided into paragraphs and has been highlighted.

Reviewer 2 Report
Comments and Suggestions for Authors
- The review article deals with studying the Role of Aloe vera on Skin and its Commensal: Contribution of 2 Acemannan in Acne cure against Propionibacterium acnes.
- The introduction section needs more details about the traditional uses and the isolated secondary metabolites from Aloe vera.
- All Figures need more improvement.
Author Response
2nd Reviewer comments
Comments and Suggestions for Authors
The review article deals with studying the Role of Aloe vera on Skin and its Commensal: Contribution of 2 Acemannan in Acne cure against Propionibacterium acnes.
- The introduction section needs more details about the traditional uses and the isolated secondary metabolites from Aloe vera.
Answer: The additional information regarding secondary metabolites isolated from aloe vera has been discussed in the manuscript in form of Table 1. The evidence of that is there after the table 1 and also in the last paragraph of section 2. The author have focused on inserting those metabolites which are largely present in aloe vera
- All Figures need more improvement.
Answer: The figures have been improved in the manuscript.
